



# The Weisweiler passive seismological network: optimised for state-of-the-art location and imaging methods

Claudia Finger[1], Marco P. Roth[2], Marco Dietl[1,2], Aileen Gotowik[1,2], Nina Engels[3], Rebecca M. Harrington[2], Brigitte Knapmeyer-Endrun[4], Klaus Reicherter[3], Thomas Oswald[5], Thomas Reinsch[1], and Erik H. Saenger[6,1,2]

[1]Fraunhofer IEG, Research Institution for Energy Infrastructures and Geothermal Systems, Bochum, Germany
[2]Ruhr-University Bochum (RUB), Germany
[3]Rheinisch-Westfälische Technische Hochschule (RWTH) Aachen University, Germany
[4]Bensberg Observatory, University of Cologne, Germany
[5]RWE Power Aktiengesellschaft, Germany
[6]Bochum University of Applied Sciences, Germany

**Correspondence:** Claudia Finger (claudia.finger@ieg.fraunhofer.de)

**Abstract.** Passive seismic datasets are a key technology for exploration and monitoring of subsurface reservoirs. Searching for alternative resources in the framework of the energy transition creates a surge for identifying as many potential sites as possible suitable for geothermal exploitation. The Lower Rhine Embayment, at the western border of North Rhine-Westphalia in Germany, is an extensional system with a very high potential for geothermal exploitation. The area experiences moderate but continuous natural seismicity. Here, we report on a passive seismic dataset recorded with 48 seismic stations centred at and around Eschweiler-Weisweiler. Background seismic noise levels are high at this site due to high levels of anthropogenic noise and thick unconsolidated sedimentary layers. The final station layout is a compromise between targeted network design and suitably quiet locations. We show that the network design allows the application of state-of-the-art methods including waveform-based source location methods and ambient noise velocity imaging methods.

## 1 Introduction

Passive seismic datasets are a key technology for exploration and monitoring of subsurface reservoirs. Subsurface seismic velocity structures, changes in the reservoir, locations of (micro-) seismic events and focal mechanisms can be determined. Results can be interpreted in terms of fault stability, in situ stress conditions and seismic hazard. In contrast to active seismic surveys, passive seismic investigations are continuous, cost-efficient but less accurate. Therefore, passive seismic recordings are especially well-suited to investigate spatio-temporal subsurface processes and can complement active seismic exploration.

The increased speed of the energy transition to renewable alternatives creates a surge for identifying as many potential sites as possible suitable for geothermal exploitation. Economic thermal energy provision from geothermal resources requires proximity to consumers and pre-existing pre-existing infrastructures. This increases environmental and societal concerns and can deteriorate the quality of passive seismic recordings through high levels of anthropogenic noise. Routinely applied methods



(i.e. picking based location schemes) are challenged and need to be supplemented or replaced by novel innovative methods capable of handling low signal-to-noise ratios Li et al. (2020).

The Lower Rhine Embayment (LRE), at the western border of North Rhine-Westphalia in Germany, is an extensional system with a very high potential for geothermal exploitation (Fritschle et al., 2021) but also moderate but continuous natural seismicity (Hinzen et al., 2021) and elevated seismic hazard (Grünthal et al., 2018). With increasing interest of local mu-

nicipalities to substitute their energy and heat production with renewable alternatives, the LRE has become a focal point for geothermal investigations (Fritschle et al., 2021). Projected active seismic exploration activities for the field laboratory planned in Eschweiler-Weisweiler are preceded by about a year of passive seismic recordings with 48 seismic stations (Figure 1). The dataset is available from doi:10.14470/MO7576467356 (last access: 03.11.2022), with a few select stations embargoed until Dec 2025.

The seismic station network design allows the application of state-of-the-art methods including waveform-based source location methods (Li et al., 2020) and ambient-noise velocity imaging methods (Rost and Thomas, 2002). Waveform-based source location methods do not rely on the identification of individual events or phases in the seismograms and are, thus, more robust when noise levels are high. Most waveform-based location methods, such as time-reverse imaging (TRI) (e.g., Finger and Saenger, 2021), use wavefield migrations and need sufficient station coverage and adequate velocity models to produce precise

locations. Ambient-noise methods can provide velocity models of reservoirs without the need for active or passive sources. They are ideally suited for sites with little to no natural seismicity and sites where dense population hinders active seismic surveys. Typically, surface waves are investigated using interferometric (e.g., Berg et al., 2018) or beamforming approaches (e.g., Löer et al., 2020). Adequate sampling in space is needed to enable robust application of ambient noise methods without aliasing. Here, we estimate the network performance using reported quality control measures for time-reverse imaging (TRI)

(Finger and Saenger, 2020) and three-component beamforming (3CB) (Löer et al., 2020) to demonstrate the applicability of the methods to our network design.

## 2 The Site

The LRE is an extensional system and represents the north-western branch of the Rhine Graben structure. Horst and Graben structures with thick unconsolidated sediments form the dominant geological structure of the LRE. Most subsurface informa-

tion is limited to the shallowest kilometre and stems from extensive lignite mining activities in the area (Vanneste et al., 2013). Major faults are oriented Northwest - Southeast with perpendicular overthrust faults (Figure 1). The region is tectonically active with moderate but continuous seismic activity (Vanneste et al., 2013).

Numerous Northwest-Southeast striking faults with an average strike direction of N130°E are intersected perpendicular by larger overthrust faults (Vanneste et al., 2013). Fault dips are estimated to be $50° - 65°$ but cannot be assigned to individual

faults due to a lack of deeper subsurface information (Vanneste et al., 2013). Mean slip rates on the normal striking faults generally do not exceed $0.07\,\mathrm{mm/yr}$ but vary between faults (Vanneste et al., 2013). The stress state at the faults has been investigated with focal mechanisms and the maximum stress is oriented towards N305°E (Hinzen, 2003); supporting the

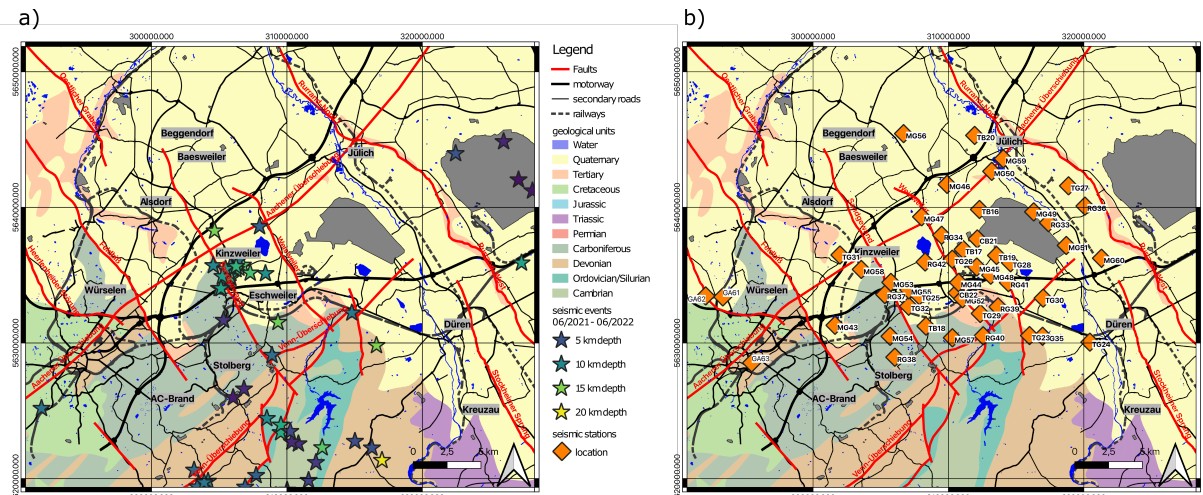

**Figure 1.** Study area in western Germany with major faults, roads and railways overlain on geological map (GD NRW, 2022). a) Seismic event locations as reported by Earthquake Observatory Bensberg University Cologne (2022) from June 2021 to June 2022. b) Station locations of the Weisweiler network are marked with squares.

overall extensional stress regime. On a local scale, the stress regime changes with depth. The shallower part (above $12\,\mathrm{km}$) is a normal faulting regime and the deeper part is a strike-slip regime (Hinzen, 2003).

Seismicity in the LRE is moderate but constant with the largest instrumentally recorded seismic event recorded on 13 April 1992 near Roermond (the Netherlands) with a magnitude of $M_b\,6.0$ (USGS, 2022). Static triggering was observed to propagate along the Roer Valley Graben System with aftershocks extending to Germany (Braunmiller et al., 1994). Dynamic triggering has been observed in our study area (Dietl, 2022). Based on the maximum fault rupture plane, the maximum possible magnitude is 7.1 (Vanneste et al., 2013), assuming complete failure at once. The estimated recurrence time for events with magnitudes

larger than 5.5 is about 211 years (Leynaud et al., 2000). The seismogenic depth is estimated to be deeper than $25\,\mathrm{km}$ (Vanneste et al., 2013) with earthquakes regularly reported deeper than $5\,\mathrm{km}$ (Hinzen et al., 2021). Due to this ongoing seismic activity, the region is categorized in the highest seismic hazard category of Germany (Grünthal et al., 2018).

Permanent seismological monitoring is done by the Earthquake Observatory Bensberg with an online bulletin published at http://www.seismo.uni-koeln.de/events/index.htm (last access: 25.08.22). The Oberservatory Bensberg operates numerous

seismic stations in western North Rhine-Westphalia and detected 88 seismic events in our study area during our investigation timeframe from June 2021 to June 2022 (Figure 1a). Extensive seismic catalogues exist for the region (Hinzen et al., 2021) that we aim to refine and enhance with the dataset presented here.

Background seismic noise levels are expected to be high at this site due to the extensive lignite mining activities, wind turbines, industry complexes and railroad and highway networks. Thick unconsolidated sediments further amplify seismic

noise (Pilz et al., 2021). The massive lignite mines in the region additionally hinder the deployment of a truly dense network. Therefore, we aim for a trade off between minimised anthropogenic noise and dense and regular inter-station spacings.



## 3 The seismic station network

The passive seismic station network should serve three main objectives:

- provide a high-quality passive seismic dataset for innovations in location and imaging methodology

- enable in-depth investigation of the seismo-tectonic processes in the deeper subsurface of an earthquake hazard region in central Europe

- enable estimation of geothermal potential and associated seismic hazard in the region

The network is centred around the upcoming drill location for an exploration well in the context of wider geophysical exploration in the region (Fritschle et al., 2021). The inter-station spacing and total extend of the network is a compromise
between dense station spacing and location and imaging capabilities in depths relevant to geothermal exploitation (2 to $4\,\mathrm{km}$ in this region (Fritschle et al., 2021)). Major faults and known seismically active parts should be covered by the stations while avoiding the vicinity of strong anthropogenic noise sources.

We use two types of methods to illustrate the network design process (this section) and estimate the network performance (section 4). The network should enable application of waveform-based source location schemes (Li et al., 2020) and frequency-
wavenumber based velocity imaging schemes (Rost and Thomas, 2002). The target depth to be analysed is slightly above and below the depth for geothermal exploitation and is thus, 1 to $5\,\mathrm{km}$. The horizontal extent of the network should enable the investigation of activity on major fault zones in the area to depths relevant to geothermal exploitation.

To design the seismic network, in a first step, the requirements of the methods are used to create an ideal network design. In a second step, the ideal locations are compared with local surroundings to exclude major anthropogenic influences. The final
step consists of finding a compromise between the ideal station layout and secure locations offered by citizens and companies.

Time-reverse imaging (TRI) is a waveform-based method for locating seismic events (Finger and Saenger, 2021). The recommended network design for TRI can be estimated roughly (Werner and Saenger, 2018) and then tested numerically (Finger and Saenger, 2020). The results are comparable to other waveform-based migration methods (e.g., Shi et al., 2022). Werner and Saenger (2018) report that the average station-spacing should be smaller than the minimum target depth. The total
aperture of the network should be at least twice the maximum target depth. Subsequently, we aimed for an inter-station spacing of $1\,\mathrm{km}$ and a total aperture of at least $10\,\mathrm{km}$. Throughout the design, the station network performance was checked numerically using the workflow proposed by Finger and Saenger (2020). The final network performance for TRI can be seen in section 4.2.

Three-component beamforming (Löer et al., 2020) is a frequency-wavenumber based method using ambient noise for deriving wave types, wavenumbers and azimuths in small time windows and for discrete frequencies. The obtained dispersion
characteristics can be used to infer shear velocity profiles. Using the general rules for wavenumber limits for beamforming applications (e.g., Löer et al., 2020) and assuming a minimal shear velocity of $v_{s,min} = 1500\,\mathrm{m/s}$ (in depths $\gtrsim 1\,\mathrm{km}$) and a maximum shear velocity of $v_{s,max} = 5000\,\mathrm{m/s}$ (Ewald et al., 2006), the required inter-station spacings $d_{min}$ and aperture





$d_{max}$ can be estimated for a frequency range of $f_{min} = 0.05\,\mathrm{Hz}$ to $f_{max} = 1\,\mathrm{Hz}$ as:

$$d_{max} = \frac{v_{s,max}}{3f_{min}} \qquad\qquad = \frac{5000\,\mathrm{m/s}}{3 \cdot 0.05\,\mathrm{Hz}} = 33.33\,\mathrm{km} \tag{1}$$

$$d_{min} = \frac{v_{s,min}}{2f_{max}} \qquad\qquad = \frac{1500\,\mathrm{m/s}}{2 \cdot 1\,\mathrm{Hz}} = 750\,\mathrm{m} \tag{2}$$

Deviating inter-station spacings or apertures change the usable frequency limits. During the design of the network, the theoretical array response (ARF) (Löer et al., 2020) is continuously checked. Results from the final ARF can be seen in section 4.1.

## 3.1 Station layout

The final station layout is a compromise between targeted network design and suitable locations offered by citizens and companies. The safety and prevention of theft of costly seismic stations is maximised by favouring locations in private gardens or fenced in industrial areas. Gaps are filled with remote locations on agricultural land, owned by RWE, where stations can be hidden in bushes at the side of fields.

The final long-term network consists of a total of 48 stations (Figure 1b) with an average inter-station spacing of $1.7\,\mathrm{km}$ and a maximum aperture of $20.6\,\mathrm{km}$ (Figure 1b). While the final layout is more irregular than ideal, it still enables high quality analyses (see section 4). Additionally, a shorter small-scale measurement was done before deployment of the larger network to estimate the general noise level.

## 3.2 Instrumentation

The instrumentation available for this study consisted of 40 stations from the geophysical instrument pool (GIPP) at the GFZ Potsdam (of which 38 were used in this study), 7 stations from the Ruhr-University Bochum (RUB) and 8 stations from the Rheinisch-Westfälisch Technische Hochschule (RWTH Aachen). In total, 27 broadband seismic stations from Nanometrics were available. The rest of the stations were short-period stations. The available instrumentation is listed in Table 1.

Station locations are named as a combination of sensor type (first letter) and owner (second letter) (see Table 1 for station codes) followed by a two digit sequential number. The GIPP and RWTH stations were equipped with batteries lasting about two months. Data were retrieved every time the battery was exchanged. The RUB stations were equipped with solar panels (TB16, TB17, CB21) or connected directly to the power outlet of home owners (TB18 - 20, CB22). The RUB stations are telemetered and continuously synchronised to data servers.

## 3.3 Deployment and Installation

At each identified station location, sensors were installed as far away as possible from roads, houses and other possible disturbances. However, due to the densely populated area and large industries, unwanted disturbances cannot be ruled out. Three stations (TB16, TB17, CB21) were deployed in the open pit lignite mine Inden in areas currently not under operation.





**Table 1.** Instrumentations available for the passive seismic network.

| Owner | no. of stations | sensor type | data recorder type | station code |
|-------|-----------------|-------------|--------------------|--------------| 
| GIPP  | 20 | Mark L-4C-3D | Digos Data-Cube 3 | MG |
|       | 10 | Nanometrics TC PH20 | Digos Data-Cube 3 | TG |
|       | 10 | Nanometrics TC PH120 | Digos Data-Cube 3 | RG |
| RUB   | 5  | Nanometrics TC PH20 | Nanometrics Centaur | TB |
|       | 2  | Nanometrics Cascadia | Nanometrics Centaur | CB |
| RWTH  | 2  | Lennartz LE-3Dlite MkII | Digos Data-Cube 3 | GA |
|       | 6  | HGS Products HG-6 4.5 Hz | Digos Data-Cube 3 | GA |

All stations were installed in shallow holes to ensure coupling to the ground. Nanometrics sensor were buried about $30\,\mathrm{cm}$ deep while short period stations were buried about $15\,\mathrm{cm}$ deep. All stations were oriented towards north with a magnetic compass. Horizontal tilt was minimised using a bubble level. Holes were backfilled with dirt or sand. Recording equipment

135 was stored in a weather proof box. A few days after installation, data quality was checked manually for unusual spikes or transients and stations were moved if necessary.

### 3.4 Recording time period

A very small short-term measurement (10 days) was performed at the center of the network in June 2021 with the 7 RUB stations (TB01 - TB05, CB06, CB07) and the 8 RWTH stations (GA08-GA15). This short term measurement served as a

140 first investigation into the expected data quality and noise levels (Gotowik, 2022) and helped design the larger network. The recorded data from this initial deployment is also available as part of the dataset.

38 of the GIPP stations (10 TG, 10 RG, 18 MG) and the 7 RUB stations (5 TB, 2 CB) comprise the main body of data. They were deployed between July and October 2021. The GIPP stations were retrieved by June 2022 while the RUB stations still continue recording as of August 2022. Data from the RUB stations will be embargoed until end of 2025 to ensure exclusive

use of these data in PhD theses at RUB. Towards the end of the investigation time period, 3 RWTH stations (GA61-GA63 in Figure 1) were deployed for a limited amount of time (March - May 2022) in the west of the network to increase coverage across the Feldbiss fault (Figure 1).

The availability of each station (excluding the small-scale array) can be seen in Figure A1. Some GIPP stations are missing data due to recorder handling issues. Due to the 2-month maintenance interval, these were only noticed at the next scheduled

maintenance. Details of each station can be found in the full station list in the Appendix (Tables A1 and B1).



## 3.5 Background noise levels

Strong anthropogenic noise and thick sedimentary layers (Vanneste et al., 2013) cause signal-to-noise ratios to be fairly low in frequency bands typically used for analysing local seismicity (Figure 2). Noticeably, noise levels are significantly lower at the south-western stations were sedimentary coverage is minimal (Figure 3). This stresses the importance of applying advanced

methods for obtaining enhanced and complete earthquake catalogues in this region. Apart from typical daily and weekly variations, the noise levels are stable in the region over multiple frequency bands (Figure 4).

The influence of the thick sedimentary layers was identified with a horizontal-to-vertical spectral ratio study (Gotowik, 2022). Resonance frequencies were found to be in the frequency range of $0.6\,\mathrm{Hz}$ to $6\,\mathrm{Hz}$. The large variability of resonance frequencies is caused by the inclined quaternary and tertiary layers (Figure 1). An elevated seismic hazard can be expected

in parts of the study with thicker sediments causing severe site amplifications at frequencies relevant to the built environment (Pilz et al., 2021).

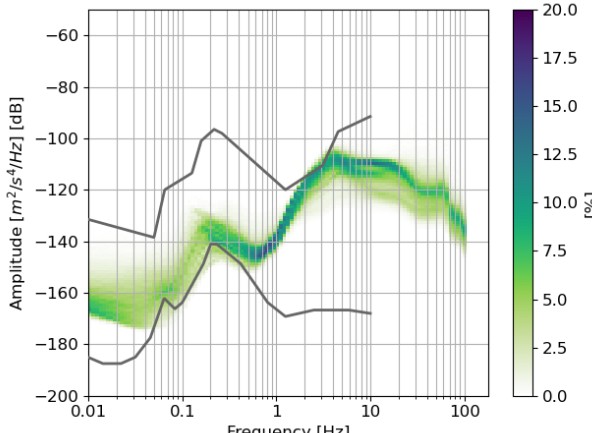

**Figure 2.** Probabilistic Power Spectral Density (PSD) created using obspy (Beyreuther et al., 2010) for the z component of station TG32 (Northeast, thicker sediments) for all recorded data. Grey lines show new low/high noise model (Peterson, 1993). This station has high noise levels.

## 4 Performance for state-of-the-art methods

Although background noise levels are elevated, local events can be recorded with adequate signal-to-noise ratios (Figure 5). A $M_L = 1.1$ event has been detected and located by Earthquake Observatory Bensberg University Cologne (2022) close to

station RG38 in May 2022. Waveforms show the heterogeneity of the network due to the complex geological structure and high noise levels.

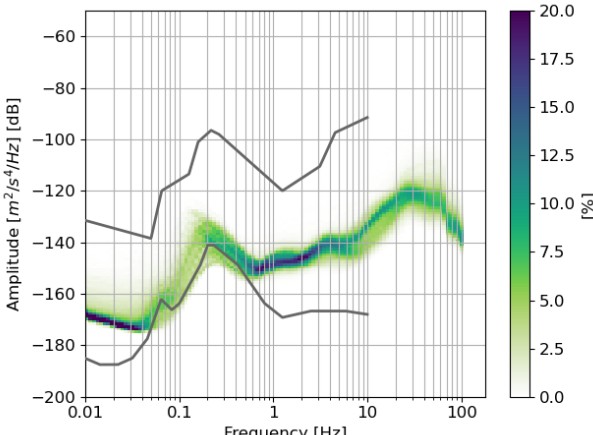

**Figure 3.** Probabilistic Power Spectral Density (PSD) created using obspy (Beyreuther et al., 2010) for the z component of station TG23 (Southwest, thinner sediments) for all recorded data. Grey lines show new low/high noise model (Peterson, 1993). This station has overall the lowest noise levels.

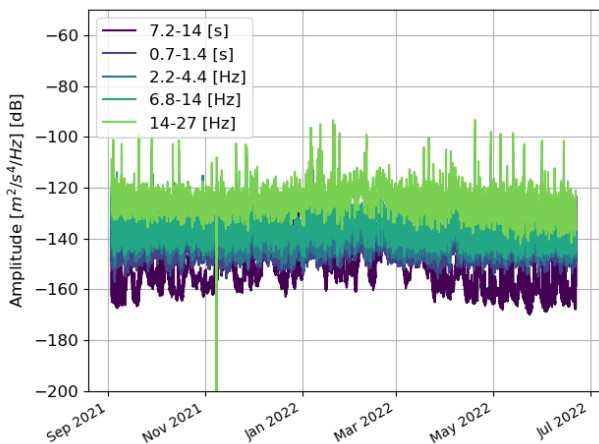

**Figure 4.** Temporal trend of PSD for selected frequencies created using obspy (Beyreuther et al., 2010) for the z component of station TG23.

## 4.1 Frequency-Wavenumber methods

Array response functions (ARF) are typically used to investigate the performance and suitability of a network for frequency-wavenumber type studies (Rost and Thomas, 2002). Using the locations of seismic stations at the surface, the wavenumber
response of the network to a wave impinging vertically from below with a specific frequency is calculated using (e.g., Löer





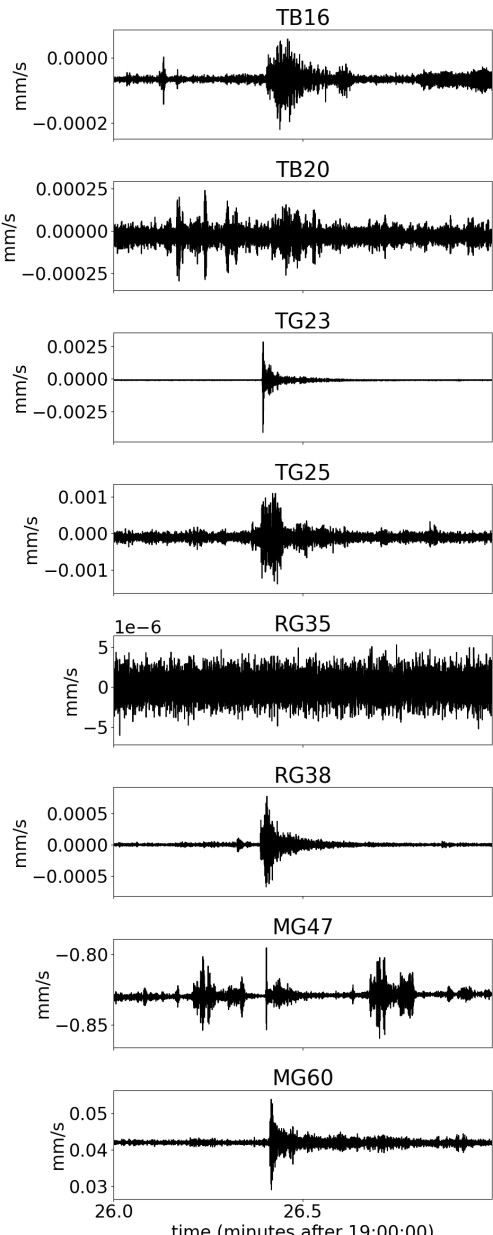

**Figure 5.** Traces on network stations recording on 05.05.2022. Traces are filtered between $10\,\mathrm{Hz}$ and $50\,\mathrm{Hz}$ around time of detected and located event in Eschweiler ($308894\,\mathrm{m}$ Easting, $5629087\,\mathrm{m}$ Northing, close to station RG38) at 05.05.2022, 19:26:21.7 with a magnitude of ML1.1 (Earthquake Observatory Bensberg University Cologne, 2022)

et al., 2020):

$$A(\mathbf{k}) = \frac{1}{M} \sum_{m=1}^{M} \exp(2\pi i (\mathbf{k} \cdot \mathbf{r}_m)), \tag{3}$$



with $\mathbf{k}$ being the wavenumber vector and $\mathbf{r}_m$ being the coordinates of $m$ stations. The final ARF for the station network (Figure 6) validates the high suitability of applying frequency-wavenumber methods to this dataset.

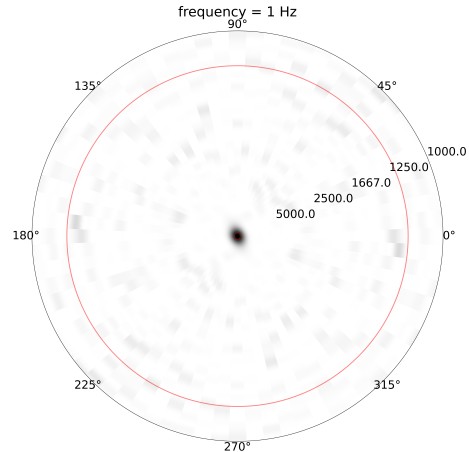

**Figure 6.** Array Response Function (equation 3) for a 1 Hz plane wave arriving from below. Radial labels represent resolvable velocity in m/s. Theoretical wavenumber limits are shown as red lines and correspond to equations 1 and 2.

Rearranging equations 1 and 2 and using the final station layout reveals a useable frequency range of $0.024\,\mathrm{Hz}$ to $1.46\,\mathrm{Hz}$ and an estimated depth sensitivity of $1\,\mathrm{km}$ to $10\,\mathrm{km}$ based on minimum and maximum resolvable wavelengths and average depth sensitivities of a fourth of the wavelength for Rayleigh waves.

### 4.2 Waveform-based source location methods

To demonstrate the applicability of waveform-based source location methods, we calculate sensitivity maps for TRI as proposed by Finger and Saenger (2020). Sensitivity maps generally estimate the impact of the station network and velocity structure on the quality of TRI results. Applied to the Weisweiler passive seismic network, sensitivity maps show the possibility for high quality results with TRI (Figure 7). To create the sensitivity maps, a total of 245 synthetic sources (circles in Figure 7), with a $3\,\mathrm{Hz}$ Ricker source wavelet, were used in a homogeneous velocity model ($v_p = 4000\,\mathrm{m/s}$, $v_s = \frac{v_p}{\sqrt{3}}$) to create synthetic seismic signals at the locations of the seismic stations. Synthetic sources are spaced regularly with $3\,\mathrm{km}$ spacing in horizontal direction and $1\,\mathrm{km}$ spacing in vertical direction. Only the $xy$-component of the moment tensor is non-zero (strike-slip sources). A three-dimensional finite-difference wave simulator (Saenger et al., 2000) with absorbing boundaries at the sides and the bottom of the model and a free surface is used to simulate the wave propagation. The model size is $24\,\mathrm{km}$ in both horizontal directions and $6\,\mathrm{km}$ in depth. The grid spacing is $20\,\mathrm{m}$ and the time step is $0.004\,\mathrm{s}$ to ensure numerical stability.

The workflow of Finger and Saenger (2020) is adapted to use slight time shifts in source origin times to allow all 245 sources to be tested with one simulation. The recorded synthetic signals at the surface stations is time reversed and normalised before back propagation. An illumination map is created to negate the influence of geometrical spreading (Finger and Saenger, 2020).





**Table 2.** Thresholds used to determine of location of synthetic sources was possible.

| quantity | threshold |
| --- | --- |
| spatial location error | $< 1300\,\mathrm{m}$ |
| normalised imaging condition amplitude | $> 0.3$ |
| timing error | $< 0.02\,\mathrm{s}$ |
| deviation from max. moment tensor component | $< 5\%$ |

The total energy density imaging condition (Saenger, 2011) is used to obtain source locations. The workflow presented in (Finger et al., 2021) is used to infer origin time and moment tensor of the source locations.

Successful source locations are defined following a set of threshold levels (Table 2). In addition to the spatial location
accuracy, the retrieved origin time and moment tensor error are considered for the determination of successful locations. Only if a source location passes all thresholds (Table 2) is the location deemed successful.

Five different depths were investigated using sensitivity maps (Figure 7). In the center of the model, all sources could be recovered. For the shallower and deepest depths, a larger number of sources at the borders of the model could not be located. Only a homogeneous velocity model is used in this investigation. As reported before (Werner and Saenger, 2018; Finger and
Saenger, 2020), a heterogeneous velocity model can cause wavefield energy to disperse away from the source locations and thus decrease the accuracy. Nevertheless, this demonstrates the usefulness of the dataset for applications of time-reverse imaging methods in specific and waveform-based methods in general.

## 5   Conclusions

Using state-of-the art knowledge about requirements of innovative location and imaging methods allowed us to design an
optimised passive seismic station network. Although some compromises had to be made between the vicinity of seismic noise sources and favouring secure locations, the station network has great potential for solving a number of open research questions.

The elevated background noise levels observed with this network provide a realistic dataset to test robustness of innovative methods in a geolgically complex area. Furthermore, this network serves as a demonstration of deploying a large passive seismic network in a densely populated area.

When planning for the application of innovative methods during the network design phase, the benefits of these methods can be reaped to overcome the challenges of the increased noise levels. Low signal-to-noise ratios are no challenge to TRI but the method needs sufficient station coverage.

All data presented here are available through the provided links. The dataset promises to be a valuable asset for the global seismological community and has potential to advance numerous methodologies while simultaneously advancing the under-
standing of subsurface structure and processes in the Lower Rhine Embayment.



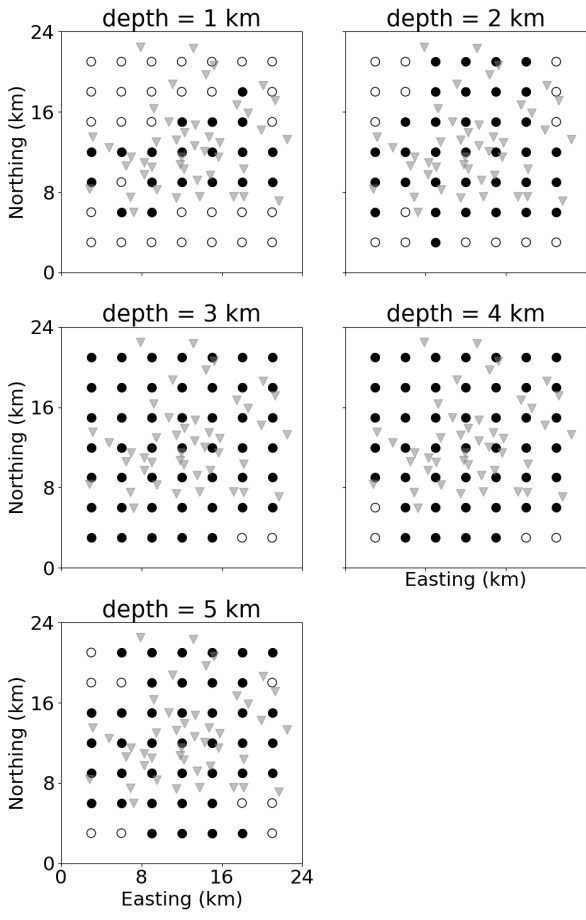

**Figure 7.** Successful locations with TRI (filled circles) appear in the center of the model and beneath the seismic stations (grey triangles). Unsuccessful locations (open circles) appear mainly at the borders and at shallow and deep depths.

## 6  Data availability

The Weisweiler passive seismic dataset presented here is available at: doi:10.14470/MO7576467356 (last access: 03.11.2022) and using the network code ZB (2021-2022) (Finger et al., 2022). The earthquake catalog of the Earthquake Observatory Bensberg at the University Cologne is available at http://www.seismo.uni-koeln.de/catalog/index.htm (last access 20.09.2022)
and waveforms are available using the network code BQ (Department of Geosciences Bensberg Observatory University of Cologne, 2016) (https://doi.org/10.7914/SN/BQ, last access 02.11.2022).





## Appendix A:  List of all stations

The complete list of stations is presented for the initial short-term deployment in Table A1 and for the long-term deployment in Table B1. Figure A1 gives an overview of available data recordings

**Table A1.** List of all stations of the short-term small-scale array including geographical location, deployment dates and retrieval dates

| station code | lat (°N) | Lon (°E) | sensor type | recorder type | date deployed | date retrieved |
|---|---|---|---|---|---|---|
| TB01 | 50.83569 | 6.31360 | Nanometrics TC PH20 | Nanometrics Centaur | 2021/06/14 | 2021/06/24 |
| TB02 | 50.83592 | 6.31332 | Nanometrics TC PH20 | Nanometrics Centaur | 2021/06/14 | 2021/06/24 |
| TB03 | 50.83543 | 6.31389 | Nanometrics TC PH20 | Nanometrics Centaur | 2021/06/14 | 2021/06/24 |
| TB04 | 50.83520 | 6.31473 | Nanometrics TC PH20 | Nanometrics Centaur | 2021/06/14 | 2021/06/24 |
| TB05 | 50.83562 | 6.31417 | Nanometrics TC PH20 | Nanometrics Centaur | 2021/06/14 | 2021/06/24 |
| CB06 | 50.83521 | 6.31332 | Nanometrics Cascadia | Nanometrics Centaur | 2021/06/14 | 2021/06/24 |
| CB07 | 50.83588 | 6.31464 | Nanometrics Cascadia | Nanometrics Centaur | 2021/06/14 | 2021/06/24 |
| GA08 | 50.83506 | 6.31365 | HGS Products HG-6 | Digos DataCube3 | 2021/06/15 | 2021/06/24 |
| GA09 | 50.83546 | 6.31253 | HGS Products HG-6 | Digos DataCube3 | 2021/06/15 | 2021/06/24 |
| GA10 | 50.83578 | 6.31259 | HGS Products HG-6 | Digos DataCube3 | 2021/06/15 | 2021/06/24 |
| GA11 | 50.83608 | 6.31327 | Lennartz LE-3Dlite MkII | Digos DataCube3 | 2021/06/15 | 2021/06/24 |
| GA12 | 50.83610 | 6.31418 | HGS Products HG-6 | Digos DataCube3 | 2021/06/15 | 2021/06/24 |
| GA13 | 50.83580 | 6.31522 | HGS Products HG-6 | Digos DataCube3 | 2021/06/15 | 2021/06/24 |
| GA14 | 50.83542 | 6.31517 | HGS Products HG-6 | Digos DataCube3 | 2021/06/15 | 2021/06/24 |
| GA15 | 50.83506 | 6.31420 | Lennartz LE-3Dlite MkII | Digos DataCube3 | 2021/06/15 | 2021/06/24 |



**Table B1.** List of all stations including geographical location, deployment dates and retrieval dates

| station code | lat (°N) | Lon (°E) | sensor type | recorder type | date deployed | date retrieved |
|---|---|---|---|---|---|---|
| TB16 | 50.87974 | 6.33164 | Nanometrics TC PH20 | Nanometrics Centaur | 2021/07/05 | - |
| TB17 | 50.85284 | 6.31603 | Nanometrics TC PH20 | Nanometrics Centaur | 2021/07/05 | - |
| TB18 | 50.80115 | 6.28004 | Nanometrics TC PH20 | Nanometrics Centaur | 2021/08/11 | - |
| TB19 | 50.84967 | 6.35123 | Nanometrics TC PH20 | Nanometrics Centaur | 2021/08/11 | - |
| TB20 | 50.92868 | 6.32374 | Nanometrics TC PH20 | Nanometrics Centaur | 2021/08/27 | - |
| CB21 | 50.86019 | 6.33048 | Nanometrics TC PH20 | Nanometrics Centaur | 2021/07/05 | - |
| CB22 | 50.82393 | 6.31125 | Nanometrics TC PH20 | Nanometrics Centaur | 2021/08/27 | - |
| TG23 | 50.79737 | 6.38880 | Nanometrics TC PH20 | Digos DataCube3 | 2021/09/03 | 2022/06/22 |
| TG24 | 50.79441 | 6.45225 | Nanometrics TC PH20 | Digos DataCube3 | 2021/09/03 | 2022/06/16 |
| TG25 | 50.82111 | 6.27194 | Nanometrics TC PH20 | Digos DataCube3 | 2021/08/11 | 2022/06/21 |
| TG26 | 50.84611 | 6.30500 | Nanometrics TC PH20 | Digos DataCube3 | 2021/08/27 | 2022/05/31 |
| TG27 | 50.89750 | 6.42417 | Nanometrics TC PH20 | Digos DataCube3 | 2021/08/19 | 2022/05/30 |
| TG28 | 50.84515 | 6.36619 | Nanometrics TC PH20 | Digos DataCube3 | 2021/08/11 | 2022/04/13 |
| TG29 | 50.81083 | 6.33640 | Nanometrics TC PH20 | Digos DataCube3 | 2021/08/11 | 2022/06/22 |
| TG30 | 50.82250 | 6.40167 | Nanometrics TC PH20 | Digos DataCube3 | 2021/08/19 | 2022/06/09 |
| TG31 | 50.84614 | 6.18709 | Nanometrics TC PH20 | Digos DataCube3 | 2021/09/08 | 2022/06/21 |
| TG32 | 50.81389 | 6.26111 | Nanometrics TC PH20 | Digos DataCube3 | 2021/09/29 | 2022/06/08 |
| RG33 | 50.87263 | 6.40453 | Nanometrics TC 120 | Digos DataCube3 | 2021/09/01 | 2022/05/30 |
| RG34 | 50.86174 | 6.29298 | Nanometrics TC 120 | Digos DataCube3 | 2021/08/27 | 2022/05/24 |
| RG35 | 50.79722 | 6.40250 | Nanometrics TC 120 | Digos DataCube3 | 2021/08/11 | 2022/05/25 |
| RG36 | 50.88445 | 6.44217 | Nanometrics TC 120 | Digos DataCube3 | 2021/08/27 | 2022/05/29 |
| RG37 | 50.82110 | 6.23567 | Nanometrics TC 120 | Digos DataCube3 | 2021/09/08 | 2022/05/09 |
| RG38 | 50.77979 | 6.24863 | Nanometrics TC 120 | Digos DataCube3 | 2021/09/10 | 2022/06/08 |
| RG39 | 50.81567 | 6.35475 | Nanometrics TC 120 | Digos DataCube3 | 2021/09/24 | 2022/06/09 |
| RG40 | 50.79611 | 6.34028 | Nanometrics TC 120 | Digos DataCube3 | 2021/09/29 | 2022/06/09 |
| RG41 | 50.83194 | 6.36472 | Nanometrics TC 120 | Digos DataCube3 | 2021/10/08 | 2022/06/22 |
| RG42 | 50.84361 | 6.27667 | Nanometrics TC 120 | Digos DataCube3 | 2021/10/08 | 2022/05/31 |
| MG43 | 50.79944 | 6.18472 | Mark L-4C-3D | Digos DataCube3 | 2021/11/23 | 2022/06/08 |
| MG44 | 50.83083 | 6.31250 | Mark L-4C-3D | Digos DataCube3 | 2021/08/27 | 2022/05/31 |
| MG45 | 50.84154 | 6.33104 | Mark L-4C-3D | Digos DataCube3 | 2021/08/25 | 2022/05/31 |
| MG46 | 50.89558 | 6.29671 | Mark L-4C-3D | Digos DataCube3 | 2021/08/25 | 2022/06/07 |
| MG47 | 50.87348 | 6.27122 | Mark L-4C-3D | Digos DataCube3 | 2021/08/19 | 2022/06/07 |
| MG48 | 50.83644 | 6.34592 | Mark L-4C-3D | Digos DataCube3 | 2021/08/25 | 2022/05/31 |

**Table B2.** *continued* List of all stations including geographical location, deployment dates and retrieval dates

| station code | lat (°N) | Lon (°E) | sensor type | recorder type | date deployed | date retrieved |
|---|---|---|---|---|---|---|
| MG49 | 50.87944 | 6.38861 | Mark L-4C-3D | Digos DataCube3 | 2021/08/19 | 2022/05/30 |
| MG50 | 50.90528 | 6.34306 | Mark L-4C-3D | Digos DataCube3 | 2021/08/19 | 2022/05/25 |
| MG51 | 50.85794 | 6.42401 | Mark L-4C-3D | Digos DataCube3 | 2021/08/25 | 2022/05/26 |
| MG52 | 50.82017 | 6.31733 | Mark L-4C-3D | Digos DataCube3 | 2021/09/10 | 2022/06/22 |
| MG53 | 50.82928 | 6.24156 | Mark L-4C-3D | Digos DataCube3 | 2021/09/10 | 2022/06/21 |
| MG54 | 50.79354 | 6.24281 | Mark L-4C-3D | Digos DataCube3 | 2021/09/08 | 2022/06/04 |
| MG55 | 50.82479 | 6.26060 | Mark L-4C-3D | Digos DataCube3 | 2021/09/10 | 2022/06/22 |
| MG56 | 50.92813 | 6.24904 | Mark L-4C-3D | Digos DataCube3 | 2021/09/08 | 2022/06/07 |
| MG57 | 50.79409 | 6.30812 | Mark L-4C-3D | Digos DataCube3 | 2021/09/24 | 2022/06/22 |
| MG58 | 50.83711 | 6.21005 | Mark L-4C-3D | Digos DataCube3 | 2021/09/24 | 2022/06/17 |
| MG59 | 50.91389 | 6.35472 | Mark L-4C-3D | Digos DataCube3 | 2021/10/08 | 2022/06/07 |
| MG60 | 50.85028 | 6.46139 | Mark L-4C-3D | Digos DataCube3 | 2021/10/08 | 2022/06/09 |
| GA61 | 50.81533 | 6.06611 | HGS Products HG-6 | Digos DataCube3 | 2022/03/25 | 2022/05/23 |
| GA62 | 50.81551 | 6.04806 | HGS Products HG-6 | Digos DataCube3 | 2022/03/25 | 2022/05/23 |
| GA63 | 50.77240 | 6.09903 | HGS Products HG-6 | Digos DataCube3 | 2022/03/25 | 2022/05/23 |

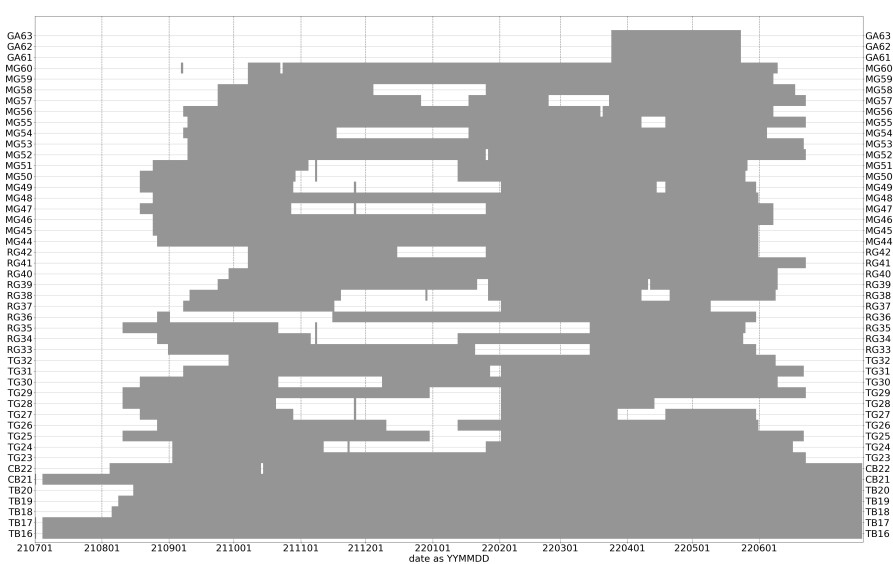

**Figure A1.** Daily data availability for the 48 stations of the passive seismic network.



*Author contributions.* CF drafted the manuscript, created figures and supervised the field deployments. CF, MPR, MD, AG, NE and RMH performed deployment of stations. CF, MPR, MD and AG did the analyses of data. CF, MPR, MD, AG, NE, RMH and BKE reviewed the manuscript. BKE, KR, TO, TR and EHS provided invaluable guidance. TR, RMH, KR and EHS procured funding.

*Competing interests.* The authors declare that there are no competing interests.

*Acknowledgements.* The authors gratefully acknowledge the support of each individual providing locations for the seismic stations. 40
seismic stations were provided by the Geophysical Instrument Pool Potsdam (GIPP) under grant agreement GIPP202110. This project has been subsidized through the Cofund GEOTHERMICA, which is supported by the European Union's HORIZON 2020 programme for research, technological development and demonstration under grant agreement No 731117. The authors gratefully acknowledge the Gauss Centre for Supercomputing e.V. (www.gauss-centre.eu, last access: 25.08.22) for funding this project by providing computing time through the John von Neumann Institute for Computing (NIC) on the GCS Supercomputer JUWELS at Jülich Supercomputing Centre (JSC). This
study was supported by the Interreg North-West Europe (Interreg NWE) Programme through the Roll-out of Deep Geothermal Energy in North-West Europe (DGE-ROLLOUT) Project (http://www.nweurope.eu/DGE-Rollout, last access: 25.08.22), NWE 892. The Interreg NWE Programme is part of the European Cohesion Policy and is financed by the European Regional Development Fund (ERDF). Activities of Fraunhofer IEG were further supported by the Federal Ministry for Economic Affairs and Energy via the subproject 'Roll-out of Deep Geothermal Energy in North-West Europe - German complementary project to Interreg North-West Europe.



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
