# Peer review of "The Weisweiler passive seismological network: optimised for state-of-the-art location and imaging methods"

_Earth System Science Data, 2022_

## Author Response (AR1)

**Author Response to Reviewer Comments on essd-2022-378**
*Original referee comment in cursive!*
**Author response in bold**

Anonymous Referee #1
*Overall, it is a readable description of the study region and aims.*

*The citation of the seismic network data set is unorthodox, however. At p. 2 line 28 it is refered to by providing the DOI. This should be cited as any other publication in the reference list, i.e. "the dataset (Finger et al., 2022) is available, with a few..." and then the details (authors, title, publisher, doi etc) in the reference list. Similarly for any other data publications with DOIs refered to in the text.*

*A few other minor flaws detract from the manuscript:*
*- p.1 "pre-exisiting" is repeated.*

*- Some careful spell checking is needed: e.g. "Oberservatory" (p.3 line 64); "extend" (not "extent"), p.8 line 79; Instrumentation (or instruments?) not "instrumentations" (caption, Table 1).*

*- Fig 2 (and perhaps elsewhere): The ObsPy community prefers "ObsPy", not "obspy".*

**Dear Anonymous Referee,**
**Thank you for taking the time to review our manuscript. As upon your suggestion, we changed the way we cite the datasets. We also carefully corrected the minor changes you suggested and once again did spell checking on the manuscript.**
**Best regards,**
**the Authors.**

Referee Andrea Rovida

**Dear Andrea Rovida,**
**Thank you for reviewing our manuscript. Your time and effort is very much appreciated. In the following we will respond to all your comments.**

*The paper presents a description of a local seismological network. The location, design, characteristics, and performance of the network during its operation time are accurately detailed. A DOI is assigned to the network and its metadata are complete and thoroughly compiled. The network metadata are easily accessible, although a few stations are embargoed, and they are encoded in the fdsn-station standardized format. The dataset is associated to a Creative Commons 4.0 licence.*

**Thank you for acknowledging our work.**

*However, the object of the paper is not a proper dataset because no original research data to be reused in further scientific research are presented, and the manuscript resembles a technical report probably not suitable for publication in ESSD.*
**We value your opinion but believe that there is high potential that this dataset will be used in future reseach studies. Designing a seismic network useable for state-of-the art methods is a research task that culminates in a passive seismic dataset. By summarizing all necessary information in this thorough description of the dataset, we aim to facilitate and trigger all kinds of research.**

*The paper is not well written and a revision of the language and terminology is necessary (some examples are indicated as "Technical corrections" below).*

**Thank you for carefully reading our manuscript. We appreciate all your comments and corrections.**

*Specific Comments:*
*The use of the term "passive seismic dataset" (lines 1, 11, 217) is very misleading, because a reader expects to find the description of a seismic (better, seismological) dataset, i.e. a compilation of waveforms/locations or other data produced by a network upon which further research can be based. Since only the data regarding the network are presented, the dataset should be referred to as a "seismic network" instead of "seismic dataset". This ambiguity is present throughout the manuscript (e.g. at lines 27-29, and especially in Section 6 – Data Availability).*
**We believe there is indeed some ambiguity. We present here the waveform dataset gathered with our seismic network. Therefore, we describe the seismic network used for acquisition of the waveform datasets and present the quality of recorded waveform data. We changed the introduction to make this clearer.**

*To stimulate the readers' interest in the paper, which is now limited and local, additional details on the recorded data and their availability and accessibility should be provided. For example, it seems that recorded waveforms are available through the GEOFON website and services, although this is not reported in the manuscript.*
**Thank you for pointing this out. We added the information in the manuscript.**

*The landing page of the DOI is hosted by the GEOFON website, which shows the presented network metadata in a clearly readable way. This is not mentioned in the manuscript but must be added and described in the "Data Availability" section.*
**We added the information about the metadata in the 'Data Availability' section.**

*Technical corrections:*
*According to ESSD's guidelines (https://www.earth-system-sciencedata. net/submission.html#manuscriptcomposition) the section dedicated to "Data availability" should precede the Conclusions*
**It seems that there was some confusion with the template. We moved the section. Thank you for your thorough comments.**
*Line 55: there are several regional European and national catalogues and papers*

*providing more accurate magnitude estimates for the 1992 Roermond earthquake than the USGS catalogue*

**Thank you for this comment. We agree that there are multiple magnitude estimates from different sources. Since all of them slightly vary, we opted to use and cite one from an independent institution that falls in the middle of the reported magnitude ranges.**

*The maps in Figure 1 are too small, in particular the lettering.*

**Thanks for this comment. We combined Figure 1a and 1b to be able to enlarge the font sizes and markers.**

*The acronyms RWE (line112), and GIPP (line 119) are not explained.*

**Thank you for pointing this out. The acronym for GIPP is given when it first appears in the manuscript: 'geophysical instrument pool Potsdam (GIPP)'. We added the explanation for the abbreviation GFZ. RWE, However, is the official company name. We added the information that RWE is an energy provider to make this clearer.**

*Here are just a few examples of errors and misuses of English grammar, syntax, and terminology but many more can be found in the text (numbers indicate the lines in the manuscript file)*
*1 and 11: a "dataset" cannot be defined as a "technology" (see also the Specific Comments)*
*17: "Economic": do you mean "Cheap"?*
*18: Pre-existing is repeated*
*20 "Novel" or "innovative"?*
*48: "Perpendicularly" instead of "perpendicular"*
*50: "Normal" is referred to a fault's type/mechanism, not a fault's strike*
*51: "Among" instead of "between"*
*53: Connect the two sentences and rephrase as "… the shallow part (above 12 km) has/shows a normal faulting regime and the deep part a strike-slip one."*
*58: What does "Based on the maximum fault rupture plane" means?*
*59: "Rupture" instead of "failure"*
*70: "Mining" instead of "mines"*
*71: "Aim at" not "aim for"*
*71: "Seismic network" instead of "seismic station network"*
*75: "an earthquake hazard region": do you mean "a high earthquake hazard region"?*
*79 "Extent" instead of "extend"*
*90: "population and industries" instead of "citizen and companies"*
*97: "is described" instead of "can be seen"*
*142 "Provide" instead of "comprise"?*
*143-144: "were still recording" instead of "still continue recording"*
*149: "following" instead of "next"*
*158: remove "frequency"*
*191: "exclude" instead of "negate"*
*198: "identified" instead of "recovered"? "Shallowest" instead of "shallower"*

**Thank you for this very thorough list. We carefully corrected all instances marked by you and additionally did another round of English language editing.**

---

## Author Response (AR2)

Author Response for Review round 2 for Manuscript 'The Weisweiler passive seismological network: optimised for state-of-the-art location and imaging methods' (ESSD-2022-378)
*Original Comments by Reviewer Andrea Rovida in italic*

*Although the revised version of the manuscript incorporates all the minor suggestions made in my previous review, I am sorry to note that it does address the main points raised about the presentation of the dataset.*

Dear Andrea Rovida,
Your comments and time are most appreciated.

*Mentioning the GEOFON website does not automatically provide "additional details on the recorded data and their availability and accessibility", as suggested in the review. I would have expected a proper description of the dataset acquired with the described network. From the manuscript, I understand that data were acquired between June 2021 and August 2022 (Section 3.4) and that a Ml 1.1 was recorded in May 2022 (Section 4 and Figure 5). As a description of the dataset, a reader would expect to know, e.g., how many waveforms were recorded, what are their characteristics and so on.*

Thank you for this constructive Comment. We have carefully considered this issue but come to the conclusion that we would remain with our level of presenting the dataset. In our opinion, the reader receives all available information in this manuscript to work with the data: The seismic station locations (Figure 1, Table A.1 and on GEOFON website), installed equipment (Table 1), type of installation (surface burial), recording durations (Table A.1 and Figure A.1), ambient seismic noise characteristics (given as power spectral density plots, Figures 2 - 4) and quality indicators for ambient noise and migration studies (given as array response function in Figure 6 and sensitivity study in Figure 7). The 'number ' of waveforms inherently results from the duration of recordings given in Table A.1 and Figure A.1 and the sampling rate of 200 Hz given directly in the header of the miniseed data itself. We added the information of the sampling rate explicity in section 3.
With this information, we feel the reader could judge the appropriateness to apply their preferred methods to this dataset. The reader is then able to download the dataset and conduct their own analysis such as deriving velocity models or earthquake catalogs.

*Findability and accessibility of the data are also poorly described and the Section "Data Availability" has to be improved with some hints about how to find the dataset and its metadata in the mentioned websites, and in which format(s) they are distributed.*

Thank you for your suggestion. The dataset can be directly downloaded from the GEOFON website through a web service and the website itself provides detailed instructions how to do so (directly linked on the landing page of our dataset). Additionally, the dataset can be downloaded in a more automatic way using the very common FDSN protocol (also detailed on the GEOFON website) using the network codes given in the manuscript. This is a standard seismological way of obtaining datasets that we feel more information is not needed at this point. Although all data on the GEOFON site are in a standardized data format, we now specify that data is archived as miniseed data in section 5 and mention that this includes metadata information for each file.

*I think all the above is what is needed to "facilitate and trigger all kinds of research", as claimed by the authors in their reply. In this respect, the Conclusions should be more specific about what are the "open research questions" and "innovative methods" generically mentioned at lines 214-215.*

We appreciate the comment to improve our Conclusions. We now specifically name a few open questions, although the complete list of possible research questions is endless.

*Apart from the few suggestions made in the previous review, the expected and needed accurate revision of the English throughout the paper has not been performed.*

We would like to refer to the Copernicus English language copy editing for further revision of the English language.

---

## Author Response (AR3)

Author's response to file upload for publication

Dear Editors, Dear Reviewers,

We gratefully appreciate your support in the reviewing process. Attached are the final production files. From the last submitted version, we updated the Affiliation list to reflect the current affiliations for all authors. We also updated the Acknowledgement section.

Best Regards,
Claudia Finger
(on behalf of all authors)